# A New, Green, Recyclable Fireproof Insulation Board for Use in Integrated Composite Structure Fire Protection Systems

**Wenxu Yang** [1,2], **B. H. Abu Bakar** [1,*], **Hussin Mamat** [3] **and Liang Gong** [4]

1   School of Civil Engineering, Engineering Campus, Universiti Sains Malaysia,
    Nibong Tebal 14300, Penang, Malaysia
2   Chongqing Dadukou District Department of Housing and Urban Rural Development,
    Chongqing 400084, China
3   School of Aerospace Engineering, Engineering Campus, Universiti Sains Malaysia,
    Nibong Tebal 14300, Penang, Malaysia
4   Department of Fire protection Engineering, Southwest Jiaotong University, Chengdu 610032, China
*   Correspondence: cebad@usm.my

**Abstract:** A fireproof insulation board can be recycled, and the raw materials used in its production are very environmentally friendly, non-toxic and non-hazardous, and bring no harm to the human body and the surrounding environment. One practical application of fireproof insulation board is in an integrated composite structural fire protection system, which is a multidimensional comprehensive structural fire protection system proposed for the combined construction of buildings with different functions, such as horizontal and vertical, to ensure overall safety in the event of a building fire. The specific new technology of this new system includes an integrated structural fire protection system composed of a 3.00 h fire insulation board, which comes from the collection of textile scraps consisting of metals and buttons removed from clothes. To prove the effectiveness of this fireproof insulation board, its parameters were collected and put into FDS (FirG Dynamics Simulator, a CFD model of fluid flow during combustion developed by the National Institute of Standards and Technology), the fire safety goals considering the safety of building and personnel were established, and fire scene design based on the statistics of fire data and building codes was generated to test the safety of evacuation. To ensure the reliability of simulation results, an on-site physical fire test was conducted with the recycled insulation board. The result shows that the function of recycled board optimizes the phased evacuation design plan of personnel and solves the design difficulties of expanding fire zones and long evacuation distances when used in warehouses. Through the innovative design of the roof opening rate set at 30% and a hole spacing of 60 m, this underground fireproof insulation board is guaranteed to possess natural smoke exhaust conditions and can be used to improve public safety areas.

**Keywords:** integrated structural fire protection system; undercover structures; new green recyclable fireproof insulation board

## 1. Introduction

For a long time, most cities in China have adopted the primitive methods of open-air stacking, natural ditch filling, and pit filling to dispose of solid waste. The treatment of waste thermal insulation materials in solid waste not only pollutes the environment, but also occupies a large amount of land for landfill, and also continuously pollutes water resources. Treatment also not only pollutes the environment, but also occupies a large amount of land for landfill, and also continuously pollutes water resources. In view of the frequent replacement of thermal insulation materials for heating bodies and pipes, a large amount of thermal insulation waste is generated. Data obtained from the China Electricity Council shows that the electric power, petroleum, chemical, steel, and other industries use a large amount of thermal insulation materials in the thermal insulation of

heating elements and pipes, and more than 95% of them are discarded or landfilled, not only causing large-scale environmental pollution, but also leading to the waste of a lot of valuable resources. Boards used as suspended ceilings, partition walls, doors, and furniture constitute an important class of materials used in housing construction [1].

Traditional thermal panels derived from polymer chemicals are problematic because of their high risk and frequency of use. Such polyurethane-based products tend to be combustible even after treatment. A different, new material for insulation board has been researched regarding flammability compared with the traditional one. Zuo and Yuan manufactured aerogels to study improvements in insulation and pressure resistance of insulation materials. According to the results, aerogels have the advantage of improving insulation performance and flame-retardant performance by reducing the thermal conductivity of insulation materials [2]. Moreno et al. conducted a study on the production of refractory cellulose boards with low density and moderate refractive strength by adding soybean protein, boric acid, and boron to wastepaper and using them as internal partitions without load; the results show the boards were reported to be suitable for use as flame-retardant building materials [3]. In the DIN 18,234 standard for testing the effects of internal fire on metal roof decks insulated with EPS, a fire protection layer made of expanded perlite offset board is recommended. Anhui Jianzhu University proved the fireproof performance of effective recycling thermosetting polyurethane plastics, whose outer side of the board is a fireproof layer composed of non-woven fabric and inorganic paste, and whose middle is rigid polyurethane foam [4].

To develop environmentally-friendly boards, cellulose-rich materials, such as recycled fibers in combination with natural-based binders, have received particular attention as construction materials due to their eco-friendly nature and favorable thermal and acoustic properties [5]. The process in Figure 1 begins with the collection of the textile scraps, their selection also consisting in removing metals and buttons from clothes. The waste is then shredded and processed through another machine that turns it into microfibers where elements are added to obtain the required fireproof properties. Finally, the material goes through a heat setting process where it is shaped into a blanket, operating by recycling and transforming textile waste into ecological panels for the thermal and acoustic insulation of buildings [6].

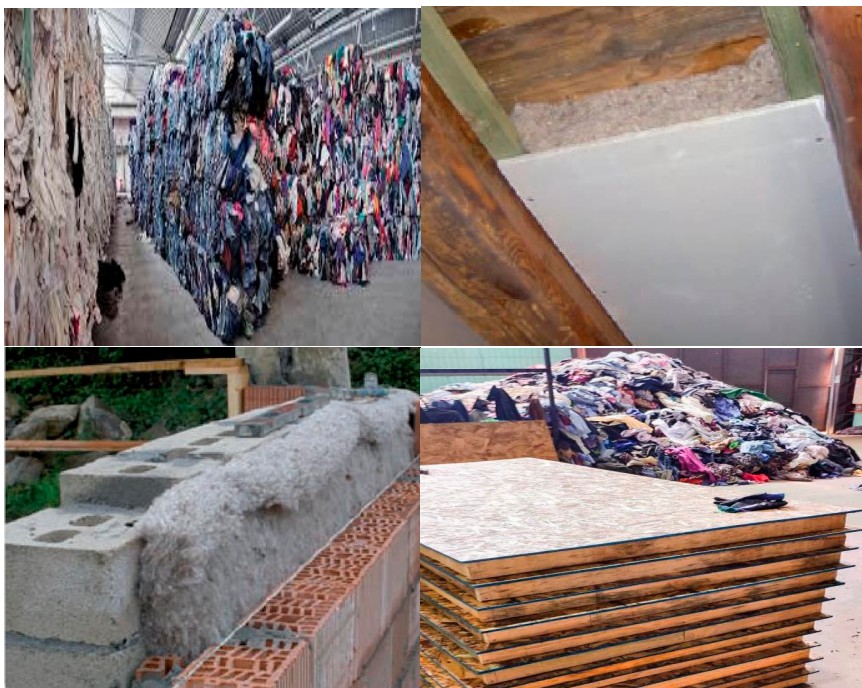

**Figure 1.** The process of recycling textile waste into ecological panels.

One practical application of recycled fireproof insulation board is in an integrated composite structural fire protection system, which is a multidimensional comprehensive structural fire protection system proposed for the combined construction of buildings with different functions, such as horizontal and vertical, to ensure overall safety in the event of a building fire. The fire resistance time of structural components is often related to factors such as the distribution of fire loads in the area and the ventilation conditions of the space. By controlling the fire separation of places with high fire danger under the cover, the fire is controlled in a certain space to prevent it from spreading out of control and causing serious damage to the building's structure. In this study, the new green recyclable fireproof insulation board was set up under cover to ensure the independence of fire rescue [7].

Undercover fireproof insulation board is different from ground fireproof insulation board in that it is completely under cover. We investigated how to ensure the safety of the undercover fireproof insulation board to ensure the safety of personnel and vehicles within fireproof insulation board during a fire so that it does not cause smoke intrusion. Combined with the terrain features, building layout and the location of the fireproof insulation board, and considering the depth of the cover and side opening conditions, a ventilation system with different opening conditions is proposed for the use of the warehouse ring fireproof insulation board. Combined with the law of smoke spread in tall spaces and theoretical calculations, a natural ventilation system with openings at the top and open sides is proposed in Figure 2, so that the new green recyclable fireproof insulation boards at different positions have certain natural ventilation and smoke exhaust capabilities to prevent smoke from affecting the safety of the lanes.

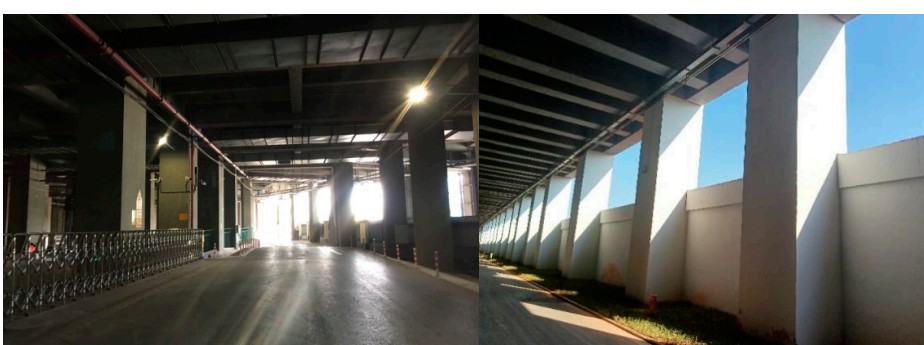

**Figure 2.** Covering the opening of new green recyclable fireproof insulation board.

Starting from the analysis of the main influencing factors affecting safety performance, we formulated fire safety strategies based on safety objectives and performance requirements.

A vertical combination of the cover was constructed, and the main fire risk was the mutual influence of fire smoke spread under the cover, as is mainly reflected in the following points:

(1) The impact of the fire under the cover and on top of the cover of a civil building mainly spreading upward through the opening of the cover plate and the edge of the cover plate.

(2) The impact of vehicle fires in relatively high-risk places such as operating warehouses and engineering garages on the safety of the overall construction's structure.

(3) The impact of fire in the building on the safety of the undercover fireproof insulation board.

(4) The impact of different building fires on the safe evacuation of personnel.

To achieve the above objectives, specific tasks shall be conducted:

(1) Collect engineering-related materials: Detailed design drawings and any supplementary materials related to the fire protection system, including the characteristics of the building itself, the operating characteristics of the building, the characteristics of users and the climate environment. For example: whether there are elderlies, chil-

dren, and disabled or other people who need help from others; whether the person is awake or asleep; etc.

(2) Determine fire safety goals: The requirements for the safety of the building itself, the requirements for protecting the safety of personnel, the requirements for protecting property, and the requirements for protecting the use of functions. For example: to ensure that people are not harmed by smoke during fire; to protect disabled people from having a safe shelter; to protect the structure of buildings from damage; etc.

(3) Determine the safety performance indicators and the conditions or indicators that should be met to achieve safety goals: For example, in order to ensure the safe evacuation of personnel, in an environment where all personnel are evacuated to—a safe area—indicators such as visibility, thermal radiation intensity, flue gas temperature and toxic gas concentration should be within safe range.

(4) Carry out fire scene design: For different assessment objectives, according to the specific conditions of the building and relevant statistical data, analyze the location of the fire, the nature of the fire, the scale of the fire, the state of the fire protection facilities, etc., and determine the most unfavorable fire situation.

(5) Evaluate the design scheme: Establish an analysis model of fire development and spread, smoke flow, and personnel evacuation; analyze whether the scheme achieves the design goals; and what aspects of the design need to be modified. For example: adding smoke detectors or automatic sprinklers, modifying ventilation characteristics, changing building materials, interior decoration and building interior decoration, etc.

## 2. Experimental Design and Method

### 2.1. Experimental Design

In this study, a simulation calculation was carried out for the safety of fireproof insulation board for use in warehouses. An opening rate of 25% of the fireproof insulation board can effectively control the smoke layer height of vehicle fire in the lane to be no less than 4m, effectively controlling the spread of smoke in large quantities. The temperature and visibility at a height of 4.0 m from the ground can ensure the safety of personnel during evacuation and fire rescue. Through the abovementioned special fire protection design, the safety design goals of covering new green recyclable fireproof insulation board and creating quasi-evacuation safety areas can be achieved.

Ensuring the safe evacuation of personnel is an important safety goal in building fire protection design. Safe evacuation of personnel means that the entire building system (including the fire protection system) can provide sufficient time for all personnel in the building to evacuate to a safe location when a fire occurs in the building, the entire evacuation process should not be subject to fire hazards. If the time it takes for the users of the building to evacuate to the safety zone (RSET) is less than the time taken for the fire to develop beyond the human tolerance limit (ASET), it indicates that the requirements for the safety of personnel are met [8]. That is, the criterion for ensuring safe evacuation is:

$$REST + TS < AEST \tag{1}$$

where REST: time required to evacuate; AEST: time when the intolerable condition of the human body begins; TS: safety margin.

The hazards of fire to people mainly come from the smoke generated by the fire, which mainly manifests in the thermal effect and toxicity of the smoke. In addition, the visibility of the smoke is also an important factor for evacuation. Therefore, when analyzing the impact of fire on evacuation, it is generally discussed in terms of temperature, concentration of toxic gases, and visibility. Table 1 shows the criteria for determining the life safety of personnel in various countries [9].

**Table 1.** Standards for human life safety determination.

| Nation | Convective Heat | Radiant Heat | Smoke Shielding |
|---|---|---|---|
| New Zealand | Smoke layer temperature $\leq 65\,°C$ (30 min) | <2.5 kW/m$^2$ (Smoke layer temperature 200 °C) | Dimming < 0.5 m$^{-1}$ visibility 2 m |
| BSI | Saturated air, exposure time > 30 min, <60 °C | Exposure time > 5 min, <2.5 kW/m$^2$ | Dimming < 0.1 m$^{-1}$ visibility 10 m |
| Australia | Saturated air, exposure time > 30 min, <60 °C | Exposure time > 5 min, <2.5 kW/m$^2$ | Dimming < 0.1 m$^{-1}$ visibility 10 m |
| Ireland | 80 °C (15 min) | 2~2.5 kW/m$^2$ | visibility 7~15m |

To predict the arrival time of danger, it is necessary to analyze the movement characteristics of the hot smoke generated by the fire in the building space under the action of the designed smoke prevention and exhaust system. This project uses the FDS field model to predict and analyze the movement of fire smoke. FDS (FirG Dynamics Simulator) is a CFD model of fluid flow during combustion developed by NIST (the National Institute of Standards and Technology). It is mainly used to analyze the movement of smoke and heat in fires [10]. The calculation of the combustion heat release rate and the radiative heat transfer by the FDS program is more accurate, and the modeling is easier. At the same time, it is closer to perfection in terms of mesh division, wall heat conduction, combustion model, and initial condition setting.

Determination of fire scale:

(1) Passage fire: For when a fire occurs in the passageway of a warehouse, we referred to the management mode of other subterranean parking lots. Since the passageway of the warehouse is clean and tidy, as the personnel are staff and cleaning personnel, there is not a large amount of combustible substances, so the possibility of fire is small. This research selects the fire of a small number of accumulated items for simulation calculation. According to the fire experiment, it is unlikely that a fire started in a dustbin would exceed 400 kw, and the fire scale of a single small item is smaller, about 200 kw~300 kw. Considering the fire test and the fire scale controlled by the staff, we determined the maximum heat release rate of fire in a warehouse passage as 2 MW.

(2) Train fire: The materials used in modern subway trains are very different from those used in previous trains. They are basically made of non-combustible or flame-retardant materials, which can reduce the generation of toxic substances and greatly improve the overall fire resistance of trains.

According to train flashover test data from the Hong Kong Mass Transit Railway's LAR line, its largest fires ranged from 5MW to 10MW, mainly due to the large number of seats included in its trains. The "Subway Design Fire Protection Standard", metro design fire protection standard GB51298-2018 article 8.2.4, stipulates that the design scale of domestic subway trains in China country is usually 7.5 MW~10.5 MW. This research conservatively uses a fast T2 fire with a maximum heat release rate of 10.5MW to characterize subway train fires to analyze the flow of fire smoke and evaluate the safety of personnel evacuation [11].

(3) Van fire on fireproof insulation board:Fire size, smoke generation, gas temperature and relevant parameters for different vehicle type were shown from Tables 2–5. According to the daily management and process requirements of the parking lot, there may be situations such as small trucks being used for the transportation of materials and the parking of vehicles.

**Table 2.** Fire size, fire perimeter and smoke generation rate for different vehicles.

| Vehicle Type | Fire Scale (MW) | Fire Perimeter (m) |
|---|---|---|
| Car | 1.5 | 5 |
| Van | 3 | 10 |
| Lorry | 10 | 15 |
| Flammable cargo | 30 | 20 |

**Table 3.** Fire scale, flue gas productivity, flue gas temperature of different vehicle types.

| Vehicle Type | Fire Scale (MW) | Equivalent Gasoline Pool Area (m$^2$) | Smoke Generation Rate (m$^3$/s) | Maximum Temperature (°C) |
|---|---|---|---|---|
| Car | 5 | 2 | 20 | 400 |
| Bus | 20 | 8 | 60 | 700 |
| Truck | | | | |

**Table 4.** Fire scale, smoke temperature, growth coefficient and burning time of different models.

| Vehicle Type | Fire Scale (MW) | Maximum Temperature (°C) | Fire Growth Factor |
|---|---|---|---|
| Car | 3–5 | 400–500 | 0.0115 |
| Bus | 15–20 | 700–800 | 0.18 |

**Table 5.** Types and sizes of fires in common areas.

| Number | Fire Type | Fire Rate | Maximum Heat Release Q (MW) | |
|---|---|---|---|---|
| | | | Sprinkler System and Effective | No Spray or When the Spray Fails |
| 1 | Solid combustible fire | Fast | 2 | - |
| 2 | Car fire | Fast | - | 7.5 |
| 3 | Van fire | Fast | - | 20.0 |

Based on the above literature, we conservatively selected a fire scale of 20 MW for the van, and selected 6 m × 2 m for the size of the van. Based on the above analysis, the fire types and scales of this project were determined as shown in Table 5.

In this study, numerical simulation analysis was carried out on fire in different areas of a parking lot passageway, a train compartment and throat area, and the entry and exit sections of fireproof insulation board. The design's fire situation in each area is summarized in the Table 6, location of fire scene is shown in Figures 3 and 4:

Table 6. Fire scene statistics.

| Fire Area | Fire Scene Number | Scene Description | Fire Scale (MW) | Fire Type | Smoke Exhaust and Supplementary Air Solutions | Set Purpose |
|---|---|---|---|---|---|---|
| New green recyclable fireproof insulation board | C1–C6 | Cargo | 20 | Fast | Natural spread | Analyze fireproof insulation board safety. |
| | Y5 | One carriage car fire | 10.5 | | Smoke exhaust failure | Analysis of personnel evacuation safety when fire blocks nearby main evacuation channels. |
| | Y6 | Solid combustible fire | 2.0 | | | |
| Engineering garage | G1 | Engineering vehicle | 20.0 | Fast | Mechanical fume extractionnatural tonic | Analyze the effect of strict control on fire of construction vehicles in the warehouse and the impact on personnel evacuation. |
| | G2-1 | | | | | Analyze the impact of smoke spread on a new green recyclable fireproof insulation board and personnel when fire shutters and window partitions in bus garages fail. |
| Throat area | H1 | One carriage car fire | 10.5 | | Natural spread | Analysis of throat area safety. |
| | H2 | | | | | Analyze throat area safety and impact on a new green recyclable fireproof insulation board. |

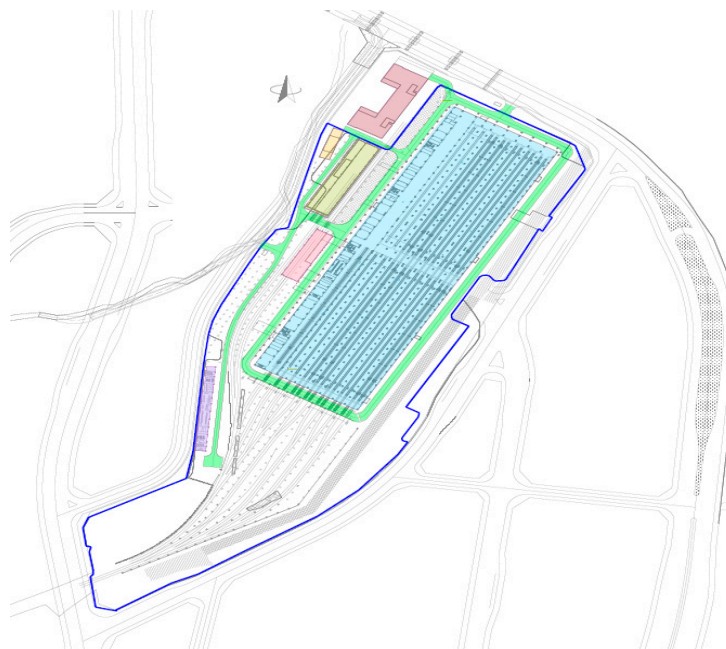

**Figure 3.** Location map of fire scene in fireproof insulation board and throat area.

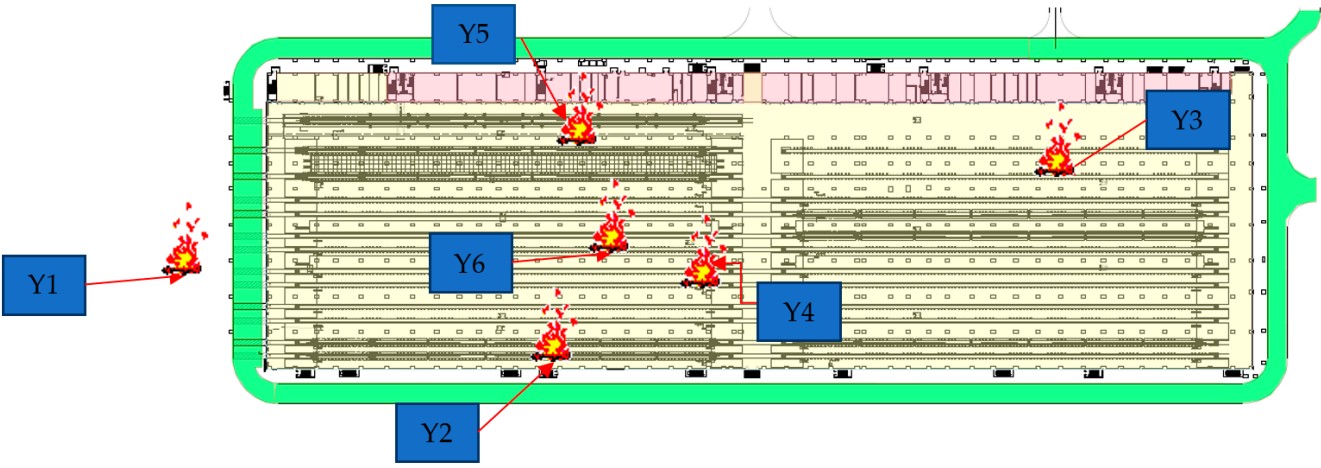

**Figure 4.** Application library fire scene location map. Smoke scenes in different location were noted from Y1 to Y6..

There are generally three methods for calculating the evacuation action time: the field simulation test measurement method, the empirical formula method, and the computer simulation method. The method of on-site simulation test measurement is mainly used for scientific research and is rarely used in engineering applications due to the limitation of funds and test conditions; evacuation prediction is carried out by manual calculation, and some countries represented by Japan mainly use this method; the computer simulation method uses computer software to simulate the dynamic process of personnel evacuation to predict the process and time of personnel evacuation movement. At present, there are mainly network models. There are two simulation models, thenetwork model and the fine mesh model. The calculation of the evacuation action time of this project uses STEPS simulation software for calculation [12]. The analysis tool used in the simulation's analysis of personnel evacuation action time is Pathfinder personnel evacuation commercial software, which is a brand-new evacuation simulator that is different from traditional software based on fluid flow calculation [13,14]. Based on computer science in the field of graphic image technology, Pathfinder achieves accurate predictions of how each individual

moves. The engineering garage is mainly used for water supply, refueling and general troubleshooting. There are few personnel. The personnel in the engineering garage are evacuated through the fireproof insulation board [15–17]. They can be evacuated to the surrounding fireproof insulation board through the surrounding evacuation doors, and then evacuated to the outside of the cover, an outdoor environment as shown in Figure 5. The distance from the evacuation door of the engineering garage to the fireproof insulation board should not be greater than 15 m.

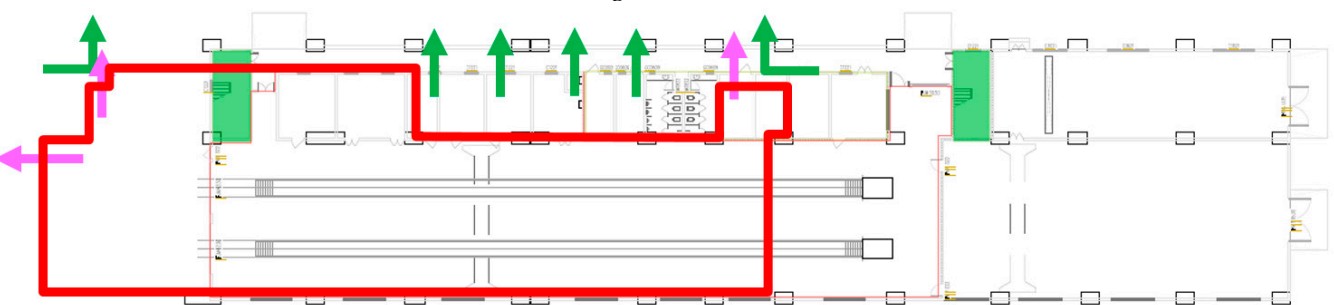

**Figure 5.** Schematic diagram of the evacuation path in an engineering garage.

The personnel in the warehouse can be evacuated through the surrounding new green recyclable fireproof insulation board. Specific evacuation path: personnel are evacuated to the fireproof insulation board through the use of the evacuation door of the warehouse, and then pass through the closed stairwell set in fireproof insulation board or use the east side slope to set up the evacuation stairs to go directly outdoors, and the personnel on the north side go directly outdoors through the main entrance and exit of the municipal road as shown in Figure 6. The interval between closed stairwells in the fireproof insulation board area should not be more than 120 m.

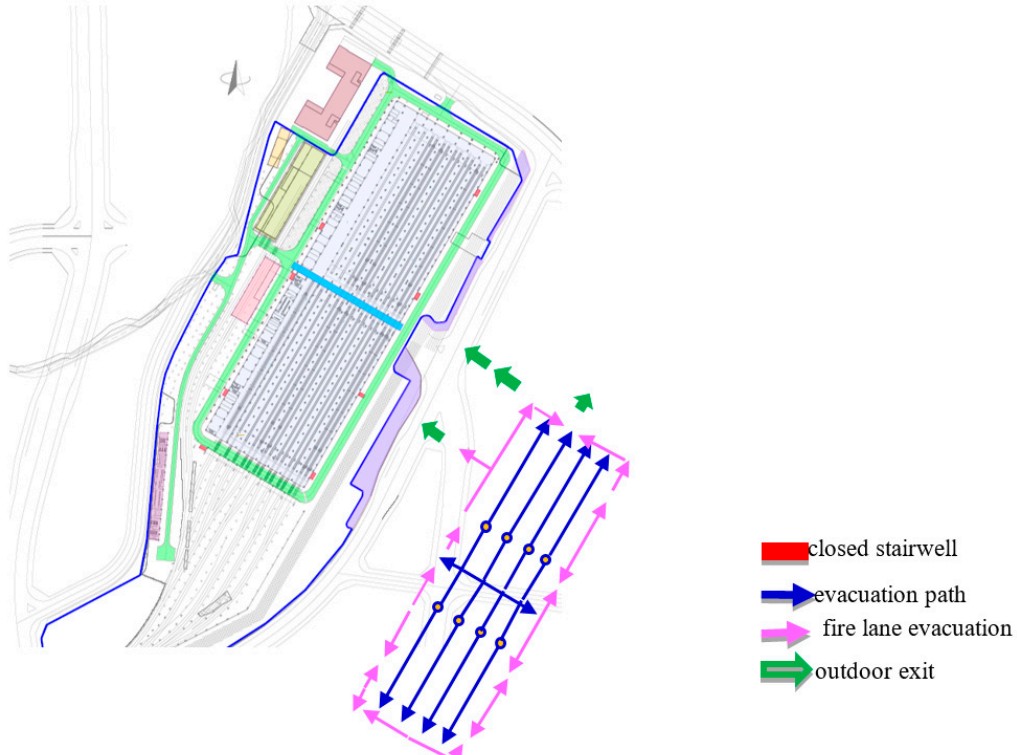

**Figure 6.** Schematic diagram of the evacuation path using the use library.

Due to the need for vehicle parking at the use library, the evacuation distance is long, and the longest evacuation distance in the use library is 115 m as shown in Figure 7:

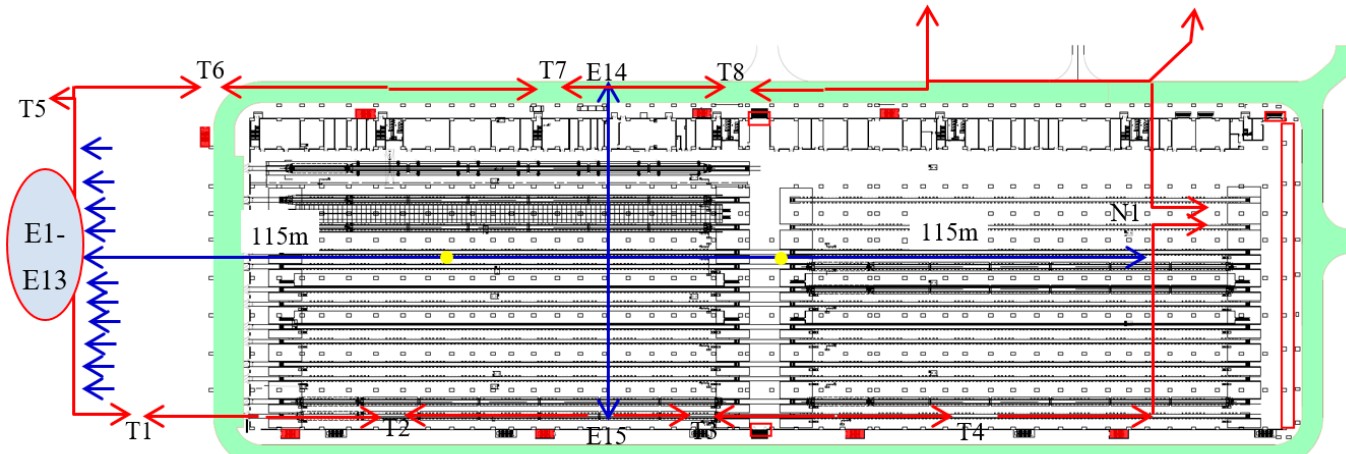

**Figure 7.** Schematic diagram of the evacuation distance of the application's library (the longest evacuation distance is 115 m).

The evacuation exits of the operating warehouse are located around the building. The specific evacuation exit location, evacuation direction, and width statistics are shown in Figure 8 and Table 7.

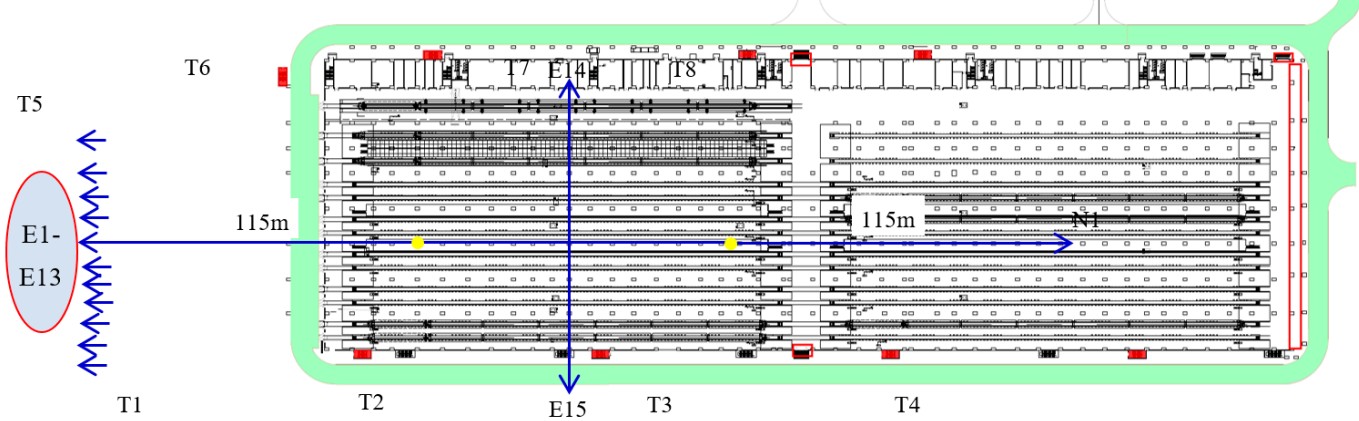

**Figure 8.** Schematic diagram of the location of the evacuation exits in the use library.

**Table 7.** Statistical diagram of the evacuation width of the application's library.

| Location | Evacuation Exit Number | Independent Evacuation Exit (m) | Total (m) |
|---|---|---|---|
| Application library | E1–E13 | 0.9 × 13 = 11.7 | 105.5 |
| | E14 | 0.9 | |
| | E15 | 0.9 | |
| | N1 | 92 | |
| Closed stairwell in fireproof insulation board | T1–T8 | 1.1 × 8 = 8.8 | 8.8 |

The proportion of each personnel classification is based on the value suggested by Simulex [18], and considering that the main functions of the building are vehicle mainte-

nance and parking functions, the types and composition of personnel in this project are shown in Table 8:

**Table 8.** Personnel type and composition.

| Personnel Type | Adult Male (%) | Adult Female (%) | Child (%) | Senior Citizens (%) |
|---|---|---|---|---|
| Garage worker | 50% | 50% | 0% | 0% |

Note: Simulex is an escape model developed by the University of Edinburgh, Scotland.

The maximum plane walking speed of each type of personnel is based on pathfinder Handbook and Simulex [19,20], and is reduced with reference to the characteristics of Chinese personnel. The determined values are shown in Table 9:

**Table 9.** Personnel speed and physical characteristics.

| Personnel Type | Average Speed (m/s) | Speed Distribution | Speed in Stairwell or Grandstand (m/s) | Body Size (Shoulder Width m × Back Thickness m × Height m) |
|---|---|---|---|---|
| Adult male | 1.3 | Normality | 0.7 | 0.5 × 0.3 × 1.7 |
| Adult female | 1.1 | | 0.6 | 0.4 × 0.25 × 1.6 |

The length of the evacuation movement time is related to parameters such as the length, width, number of personnel, and travel speed of the evacuation passage in the building. The speed of human travel is related to population density, age, and flexibility. When the density of people is less than 0.5 people/m$^2$, the traveling speed of the crowd on level ground can reach 70m/min without crowding, and the speed of descending the stairs can reach 51–63 m/min [21–23]. On the contrary, when the population density is greater than 3.5 people/m$^2$, the crowd is very crowded and can hardly move. In addition, referring to the provisions of 5.5.16 of the "*Code for Fire Protection of Building Design*" (GB50016-2014 (2018 edition)), the personnel flow per share on flat sloped ground is 43 people/min, and the personnel flow per share on stepped ground is 37 people/min. Thus, the flow of people per minute per meter's widths (i.e., outflow coefficient) are 43/0.55 = 78 (people) and 37/0.55 = 67 (people), respectively [24–27]. The general principle of evacuation scene design is to find out the most unfavorable situation for the safe evacuation of personnel after a fire occurs. The personnel evacuation scenarios of this project are set as shown in Tables 10 and 11. Figures 9 and 10 show the evacuation scenario diagram and fire scene location:

**Table 10.** Evacuation scenario table.

| Evacuation Scenario | Location | Fire Scene | Exits and Stairs |
|---|---|---|---|
| S1 | | Y1 | Block the evacuation passages and nearby exits on both sides of the train fire, while other passages and exits are not blocked. |
| S2 | | Y2 | Block the exit of the evacuation passage near the train fire, other passages and exits are not blocked. |
| S3 | Application library | Y3 | Block the fire channel, other channels and outlets are not blocked. |
| S4 | | Y4 | Block the evacuation passages and middle passages on both sides of the train fire, other passages and exits are not blocked. |
| S5 | | Y5 | Block the evacuation passages and exits on both sides of the train fire, other passages and exits are not blocked. |

**Table 11.** Smoke scene Y1 parameters.

| Parameter Category | Parameters | | | |
|---|---|---|---|---|
| Scene set | Scene number | Y1 | Location | Using a garage train fire |
| Outside wind | No wind | Speed | 0 | - |
| Fire source set | Fire scale (MW) | 10.5 | Fire growth pattern | Fast t2 fire |
| | Soot generation | 0.05 | Fire growth rate $(kW/s^2)$ | 0.04689 |
| Exhaust system | Smoke exhaust method | Mechanical fume extraction failure | | |
| Calculated parameters | Ambient temperature (°C) | 20 | Simulation period (s) | 1800 |
| | Turbulence model | Large eddy model | Environment humidity | 40% |
| Calculation Software | FDS | | | |

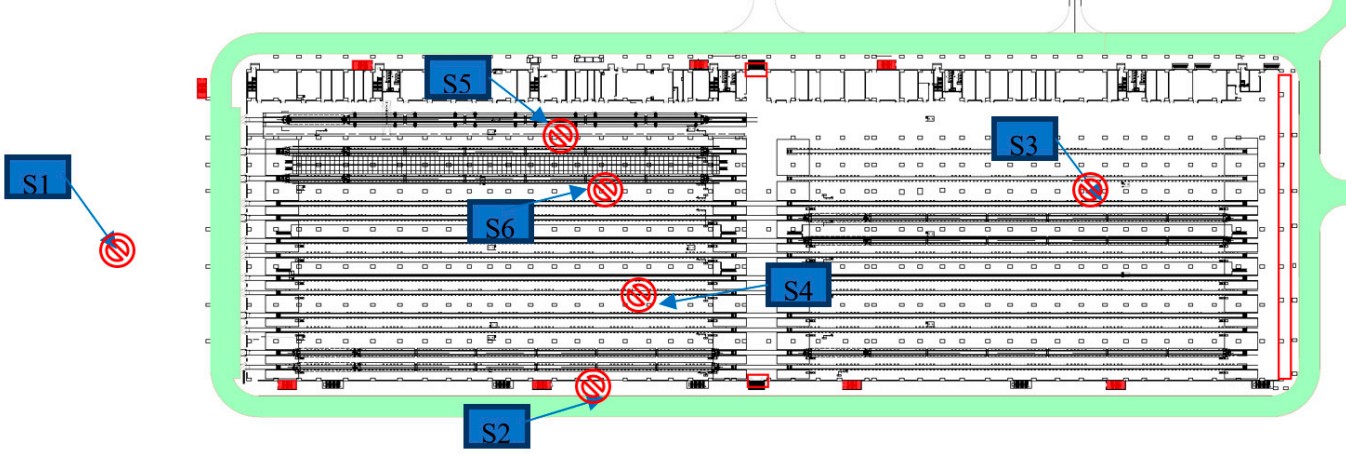

**Figure 9.** Evacuation scenario diagram.

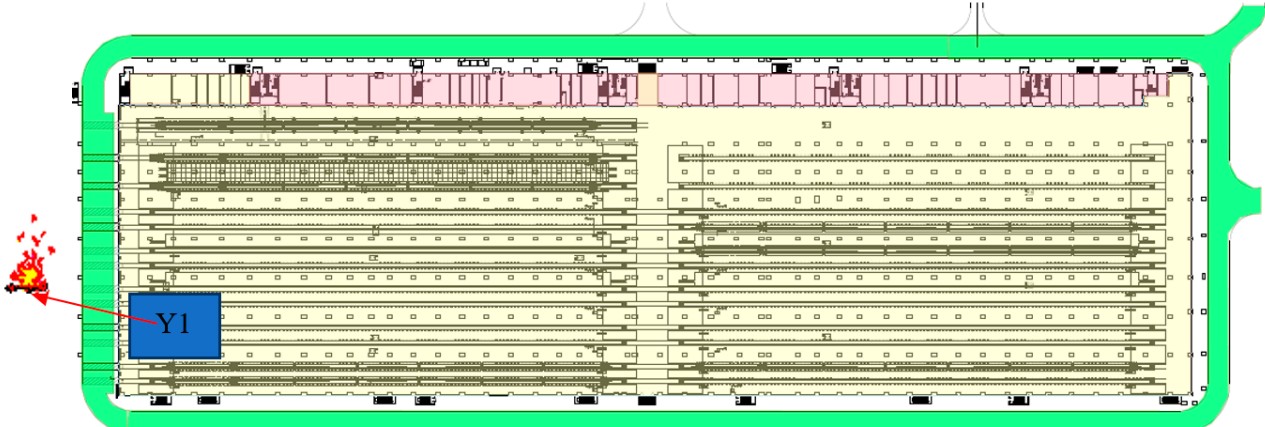

**Figure 10.** Location map of fire scene using library Y1.

### 2.2. On-site Fire Physical Test

The purpose of the test is to study the fire smoke spread and control law of the rail vehicle base under the full-scale fire experimental conditions, and to evaluate the effectiveness of the fireproof insulation board. The specific purposes are as follows:

(1) When a fire occurs in the garage area and rail area, verify the feasibility of the fireproof insulation board as a "quasi-safety area";

(2)    The feasibility of natural smoke exhaust when a fire occurs in the fireproof insulation board.

In this study, a hot smoke test was conducted on an underground fireproof insulation board. The fire source system was used to generate hot flue gas with preset fire power, including burner, flue gas generator and flue gas generating box, and protective device. Since the fire source was arranged in the most unfavorable position of the garage area, the track area and the fireproof insulation board, this point was the most unfavorable to the evacuation of personnel and posed the greatest danger to personnel. Therefore, each relevant test point was arranged near the fire source, the evacuation path and the smoke outlet, so as to obtain the distribution of relevant risk indicators. The middle position of two adjacent lighting ventilation shafts (that is, the most unfavorable point for natural smoke exhaust; the distance from the lighting ventilation shaft was 30 m) was selected as the smoke emission test point, the fire source was a diesel pan of 1.82 m$^2$, the fire heat release rate was 2.5 MW, and the smoke exhaust test was carried out under two working conditions (the height of the oil pan from the ground was 1.3 m and 0.1 m, respectively), and the combustion time was about 12 min each time [28].

## 3. Results and Discussion

Consider the unfavorable situation of a 10.5 MW train fire occurring on the south side of the warehouse, and the mechanical smoke exhaust system failing; after the fire, the flue gas rises to the ceiling of the use storehouse, settles continuously, and generates a ceiling jet to the surrounding area in Table 12. Due to the large area and high net height of the use storehouse, and because there is no separation measure on the north side, the effect of diluting the flue gas is better. The flue gas is continuously replenished and entrained by fresh air, and spreads to the entire application warehouse by thermal buoyancy, and a large amount of flue gas is discharged outdoors through the north cover. In Figure 11, under the action of hot air pressure, a small amount of smoke passes through the hole; spread along the cover plate to the throat area, the smoke can be discharged through the throat area top hole. It extends about 150 m to the throat region, and later stage smoke concentration is lower. At this length, the natural flue gas emission can be satisfied by increasing the current net flow area. By 1800 s, the visibility of other areas at a distance of 4 m from the ground is greater than 10 m except near the fire source; by 1800 s, except for the local area near the fire source and the northern corner, in other areas the temperature at 4 m from the ground is less than 60 °C in Figure 12.

In this working condition, the fire danger advent time (ASET) at a distance of 4 m from the ground in the warehouse is greater than 1800 s, the fire danger advent time (ASET) at a distance of 6 m from the ground is 680 s, and the time required for the safe evacuation of personnel on the 3.5/3.6 m platform (RSET) is 413.5 s; ASET > RSET, the personnel in the warehouse can be evacuated safely.

A train in the warehouse is used for fire, and the smoke is mainly discharged to the outside through the edge of the cover plate on the north side. The warehouse has a large area and a high net height, and the effect of storing and diluting smoke is better. In the case of smoke failure, a large amount of smoke settles in the ceiling. Although it does not affect the safe evacuation of personnel in the use warehouse, the safety margin of the time required for the safe evacuation of personnel is also short.

**Table 12.** Smoke scene Y1 modeling.

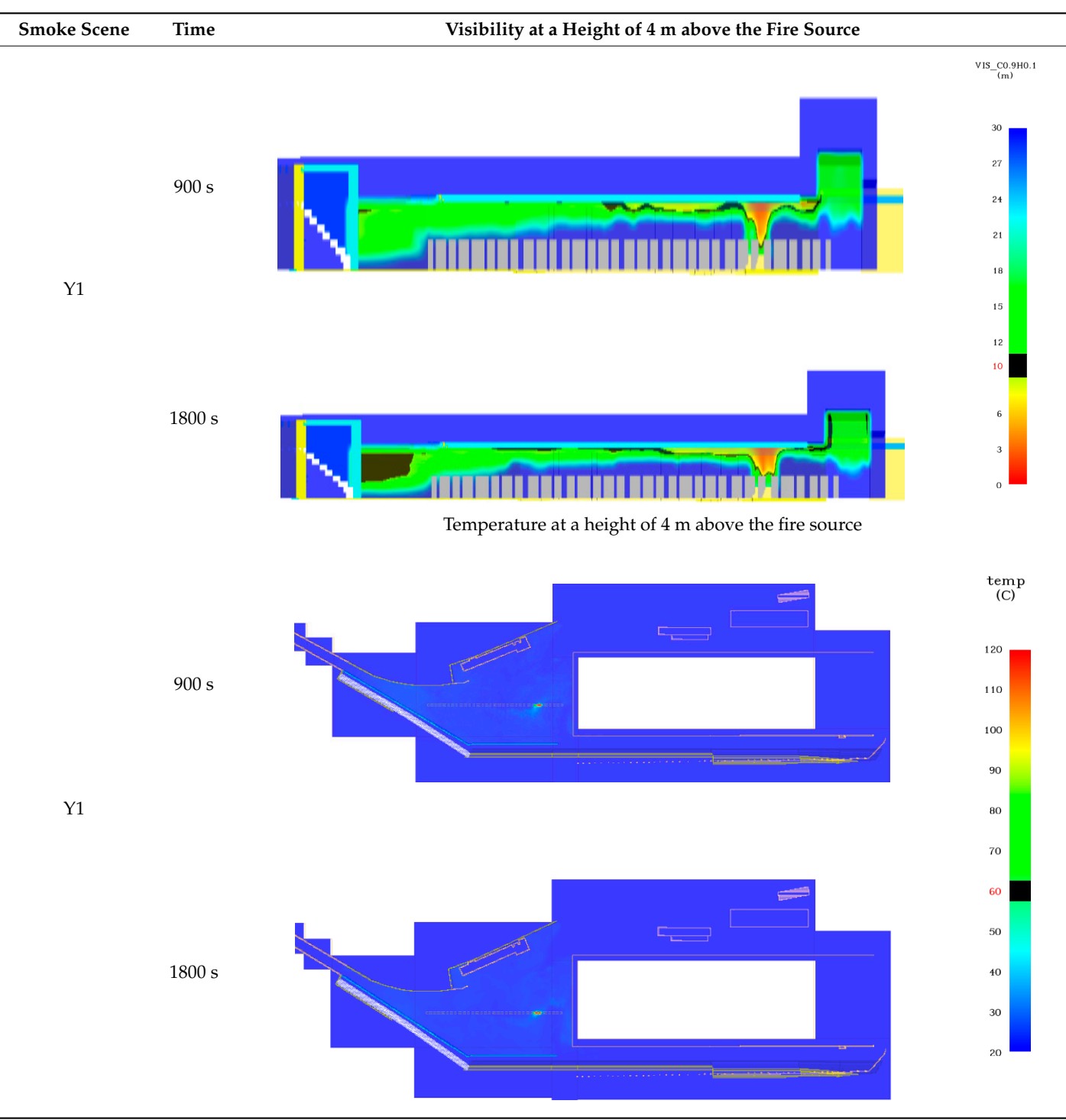

| Smoke Scene | Time | Visibility at a Height of 4 m above the Fire Source |
|---|---|---|
| Y1 | 900 s | |
| | 1800 s | |

Temperature at a height of 4 m above the fire source

| | 900 s | |
|---|---|---|
| Y1 | 1800 s | |

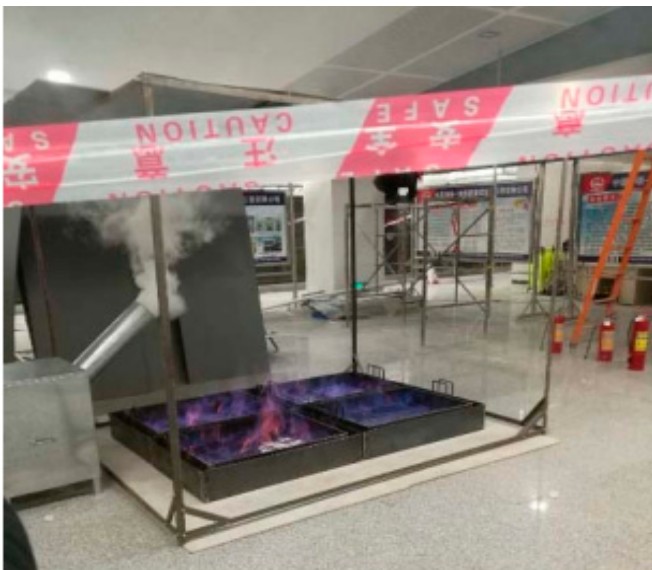

**Figure 11.** Diesel Pan of 1.82 m$^2$.

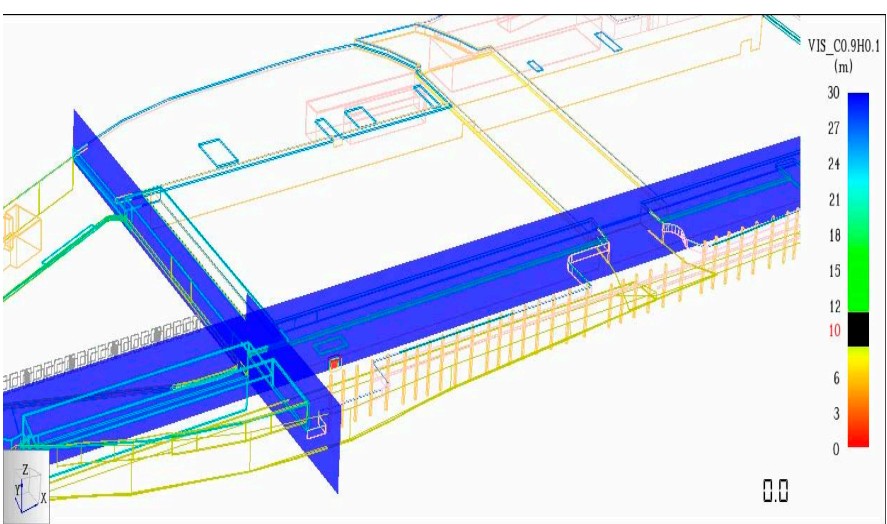

**Figure 12.** Throat area smoke spread.

Through the simulation analysis of the smoke and evacuation in the operation warehouse and the engineering garage, it can be concluded that when a fire occurs, the personnel in the operation warehouse can be evacuated to the outdoor safe area within 505 s, and the personnel in the engineering garage can be evacuated to the outdoor safe area for the longest time. They can be evacuated to an outdoor safe area within 340 s, and the time for fire danger is greater than the evacuation time for personnel. Under the existing conditions, this meets the safety of personnel requirement in the application warehouse and engineering garage in Figure 13. It can be seen from the simulation that with the passage of time, the openings at the top of the throat area and the openings on the entry and exit lines can effectively control the spread of fire smoke from the train, and the openings at the top can smoothly exhaust smoke. The ground in the throat area is 4.0 m away. The temperature and visibility at this height can ensure the safety of personnel evacuation and fire rescue. And the smoke layer on the fireproof insulation board that reaches the substation can be basically stabilized at about 4 m from the ground, and the smoke does not spread to other new green recyclable fireproof insulation boards and tunnels.

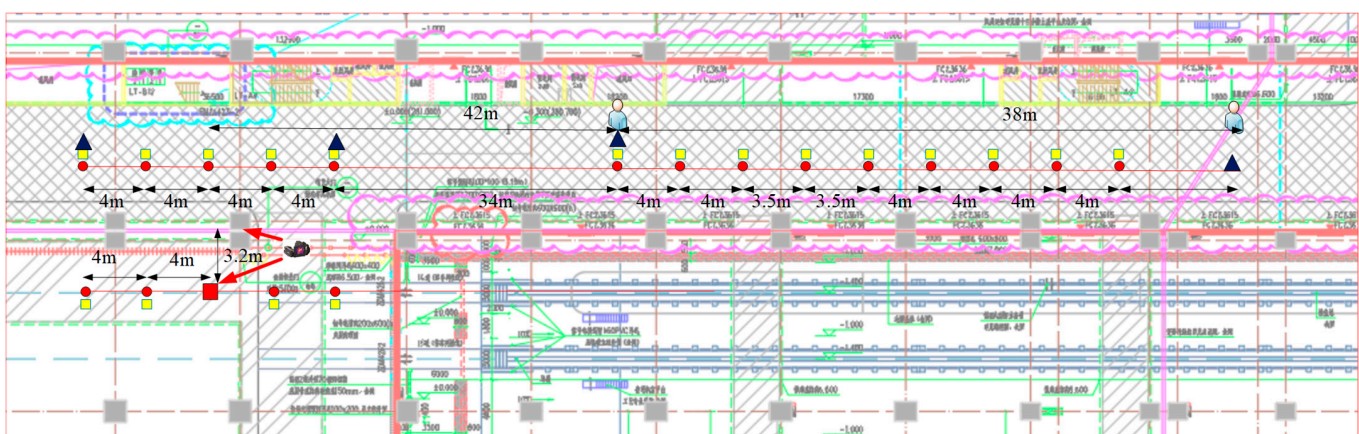

**Figure 13.** Measuring points distribution.

For the onsite fire physical test, a measurement system is used to obtain the index parameters of the on-site hot smoke test. The main measurement units include a temperature measurement unit, a gas concentration measurement unit, an airflow velocity measurement unit, an image information acquisition and display unit, a thermal image measurement unit, and a smoke layer height indication unit. During the whole test process, the spread and accumulation of hot smoke in the underground fireproof insulation board was visually simulated.

The results of on-site fire physical test measuring points show the following:

(1) When a fire occurs in the fireproof insulation board, the generated hot smoke can rise and gather on the roof of the lane in the form of axisymmetric smoke plumes, spread against the roof, and finally spread to the outdoor space through the roof opening;

(2) The visibility at the elevations of 2.3 m, 4 m, and 5 m from the ground in the underground fireproof insulation board are all higher than 25 m;

(3) Due to the natural smoke exhaust effect of the ventilation shaft, the smoke layer is basically stable and maintained at a certain height (5.5 m above the ground);

(4) The spread of smoke is always limited between two ventilation shafts, and natural smoke exhaust is achieved through the ventilation shafts.

## 4. Conclusions

On-site fire physical test and computer simulation calculation analysis of flue gas show that the design of the opening on the top of the underground glass wool fireproof insulation board can effectively discharge the flue gas generated during the test to the outside (that is, to meet the requirements of natural smoke exhaust), and keep the smoke layer under the smoke layer. The new technology optimizes the phased evacuation design plan for personnel and solves the design difficulties of expanding fire zones and long evacuation distances in the use of warehouses. Through the innovative design of setting the roof opening rate at 30% and a hole spacing of 60 m, the underground fireproof insulation board is guaranteed to have natural smoke exhaust conditions and can be used as a public safety area.

Due to the different natural ventilation and smoke exhaust capacity of fire lanes in different locations, it is necessary to further consider the impact of smoke on lane safety. Based on the rule of smoke spread in high space with theoretical calculation, all kinds of natural ventilation systems have to be taken into account, such as opening at the top and opening at the side. As the entity fire simulation does not have the relevant conditions, a theoretical formula calculation for ventilation and smoke exhaust will be provided.

**Author Contributions:** Conceptualization, W.Y., B.H.A.B. and L.G.; methodology, W.Y. and L.G.; software, W.Y.; formal analysis, H.M.; investigation, W.Y. and L.G.; resources, W.Y.; writing—original draft preparation, W.Y. and B.H.A.B.; writing—review and editing, W.Y. and B.H.A.B.; supervision, B.H.A.B.; project administration, W.Y. and B.H.A.B. All authors have read and agreed to the published version of the manuscript.

**Funding:** This research received no external funding.

**Institutional Review Board Statement:** Not applicable.

**Informed Consent Statement:** Not applicable.

**Data Availability Statement:** Data available on request from the authors.

**Conflicts of Interest:** The authors declare that they have no known competing financial interests or personal relationships that could have appeared to influence the work reported in this paper.

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
