# Peer review of "A New, Green, Recyclable Fireproof Insulation Board for Use in Integrated Composite Structure Fire Protection Systems"

_fire, doi:10.3390/fire5060203_

Round 1
Reviewer 1 Report
The manuscript presents the fireproof performance of new green recyclable fireproof insulation board via fire modeling and physical fire test method. The aim is clear and well written. The method fire and evacuation model used in this research is valid. I therefore recommend this paper to be published. And it is better if the authors improve the following mentioned remarks before submitting the final version.
1. In the introduction, the authors are encouraged to add some recent references on recycled fireproof insulation board.
2. Authors should include an elaborate discussion explaining the findings regarding how the results prove the fire performance of the material in details;
3. The research covers lots of work. It is suggested to add research limitations and further study.
4. More details for fire simulation modelling parameters should be added, such as reaction equations.
Reviewer 2 Report
1、 Fire safety goals considering the safety of building and personnel are established, fire scene design based on the statistics of fire data and building codes is generated to test the safety of evacuation. Reliability of simulation results are authenticated;
2、the underlying approach in the paper is well known and understood, and is the correct approach to pursue when inserting these types of new materials into buildings;
3、A more appropriate analysis here would be studies on the flammability of the new insulation board vs. the older ones, and comparison/contrasts between the two and how they do/do not change flame spread/smoke release.
4.references for the fire sizes in the table should be included
Reviewer 3 Report
1. The novelty could be summarized in two aspects:recycled panels and fire protection system combined with different technology,which seem to be practical and feasible,meanwhile fix the issue of waste resources,will be contribution to environment if implemented.
2. Data collection、fire and evacuation modelling fit the requirement of FDS simulation,a real fire test makes the design more persuasive.
3.More data analysis for the real fire test could better support final conclusion.
